# DISCO: Mitigating Bias in Deep Learning with Conditional DIStance COrrelation

## Abstract

During prediction tasks, models can use any signal they receive to come up with the final answer - including signals that are causally irrelevant. When predicting objects from images, for example, the lighting conditions could be correlated to different targets through selection bias, and an oblivious model might use these signals as shortcuts to discern between various objects. A predictor that uses lighting conditions instead of real object-specific details is obviously undesirable. To address this challenge, we introduce a standard anti-causal prediction model (SAM) that creates a causal framework for analyzing the information pathways influencing our predictor in anti-causal settings. We demonstrate that a classifier satisfying a specific conditional independence criterion will focus solely on the direct causal path from label to image, being counterfactually invariant to the remaining variables. Finally, we propose DISCO, a novel regularization strategy that uses conditional distance correlation to optimize for conditional independence in regression tasks. We can show that DISCO achieves competitive results in different bias mitigation experiments, deeming it a valid alternative to classical kernel-based methods.

## 1 Introduction

Deep neural networks can achieve compelling results in complex prediction tasks by learning representations from raw input data LeCun et al. (2015). However, parts of the input data might be unstable, biased, and unreliable. Consider the classification of images into cats and dogs. Suppose most cat images are captured outdoors, whereas dog images are mainly taken indoors. A naive model might capture the background as a salient feature, as it presents a strong signal across the training dataset. Thus, rather than learning to discriminate between dogs and cats based on their intrinsic characteristics, the model might erroneously prioritize background cues, rendering these models unreliable. Preventing models from picking up such unreliable or undesirable cues has been discussed in the literature from different perspectives:

*Domain Generalization* (DG) asks if we can make a prediction model reliable in changing domains and data distributions. In our cats and dogs example, one might ask how much our trained model will deteriorate if employed in a new domain where cats and dogs are randomly placed outside and inside. Arjovsky et al. (2019) and Sagawa et al. (2019) deal with samples from different domains with different data distributions and propose methods to make a model reliable in all of them. Instead of only having data from different domains, some approaches explicitly model distribution changes in certain attributes and assume one has direct access to these variables while making a model robust to such changes (Makar et al., 2022; Makar & D'Amour, 2022; Zheng & Makar, 2022; Jiang & Veitch, 2022; Wang & Veitch, 2022; Pogodin et al., 2022; Mouli & Ribeiro, 2022; Mahajan et al., 2021; Eastwood et al., 2022; Heinze-Deml & Meinshausen, 2021). Lastly, as a an attempt to unify DG methods, Kaur et al. (2022) model the domains/environments and additional bias attributes, providing causal graphs for relevant settings in many DG tasks.

*Fairness* attempts to avoid the use of unfair information in arbitrary machine-learning tasks (Mehrabi et al., 2021; Barocas et al., 2023). In our example, we may consider it unfair that indoor pictures are classified as dogs, independent of the photographed animal. A more common fairness scenario is to analyze the potential discrimination of hiring practices with respect to protected attributes (sex and race) by computing observational criteria (e.g., demographic parity, equal opportunity) (Barocas et al.,

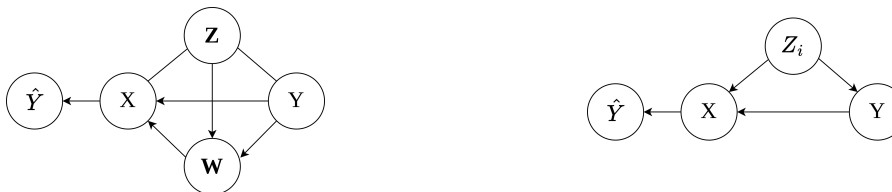

(a) Standard Anti-Causal Model (SAM)  (b) A simple instance of SAM

Figure 1: (a) The Standard Anti-Causal Model (SAM) considers the anti-causal prediction of the target $Y$ from input $X$ only, where $\hat{Y}$ is the prediction by a fixed model represented as a random variable. $\mathbf{W}$ is a set of variables that only permits mediated effects from $Y$ to $X$. $\mathbf{Z}$ contains variables lying on non-causal, open paths from $Y$ to $X$, denoted by edges without arrowheads in this case. Variables in $\mathbf{Z}$ can have any connections inside their set, and variables in $\mathbf{W}$ do not have any other arrows between them. See App.A for more details. (b) illustrates a simple instance of SAM.

2023). More advanced approaches integrate a causal graph to check for unfair information flow in the classification. Kusner et al. (2017) ask the counterfactual question if the decision for an individual would have changed if we changed a protected attribute, keeping the rest fixed. Another approach is to use path-specific fairness analysis in different prediction contexts  (Kilbertus et al., 2017; Chiappa, 2019; Chikahara et al., 2021; Zhang & Bareinboim, 2018).  Lastly, Plecko & Bareinboim (2022) provide a comprehensive study by defining a standard unifying causal graph that encodes their main assumptions, rendering it applicable to many scenarios in causal fairness analysis.

*Causal Inference* is the study of cause-effect relationships between variables and events (Pearl, 2009). It provides us a language to describe statistics beyond observations and correlations and helps us in estimating causal effects. Furthermore, with varying complexity and depth (e.g. allowing for temporal changes, heterogeneous effects, ...), causal inference usually assumes the existence of stable causal mechanisms. Returning to our cat and dog example, we can categorize it as an anti-causal prediction task (Schölkopf et al., 2012), since the labels cause the input image as illustrated in Fig. 1. In general, we assume that a dog always looks like a dog and there is a stable mechanism that makes it discernible from other animals, independent of varying lighting conditions, the animal's surroundings, or the camera lens used to capture the image. We can use this kind of causal language and analysis to make a model rely solely on task-relevant features instead of being fooled by spurious or environment-specific attributes. For example, Veitch et al. (2021); Quinzan et al. (2022), model the prediction task as a causal graph and define *counterfactual invariance* as a predictor's stability to counterfactual changes of distracting attributes. Given a single instance of a dog (Veitch et al., 2021) (or a set of dogs (Quinzan et al., 2022)), they ask whether the model's prediction changed if the background was replaced with another one.

We see increased adaptation of causal techniques in fairness and DG and propose to unify different views from before in the general anti-causal prediction setting. While (Veitch et al., 2021; Quinzan et al., 2022) did solid pivotal work by coining terms such as counterfactual invariance and arriving at useful observational criteria to achieve invariance, they used different causal settings binding their statements to concrete graphical setups. Especially their final statements in invariance criteria had significant discrepancies and their counterfactual statements where on different levels.  We want to provide a unifying approach by employing a more general and flexible graph, and treat our counterfactual statements very explicitly.

**Contributions:** Our contributions are twofold. First, we introduce the *standard anti-causal model (SAM)*, which extends the standard fairness model (Zhang & Bareinboim, 2018; Plecko & Bareinboim, 2022) to be permissive for a broader set of causal relationships in the context of anti-causal prediction. Figs. 1 and A.1 present the causal graph of SAM, where we assume that the effect of the target $Y$ on $X$ is the only relevant/desirable one for the prediction task. We derive in Sec. 2 that a predictor that satisfies the conditional independence $\hat{Y} \perp \mathbf{W}, \mathbf{Z} \mid Y$ will not leak any unwanted spurious information over $\mathbf{Z}$ or any irrelevant indirect information over $\mathbf{W}$. We also prove that the counterfactual direct effect from $Y$ to $\hat{Y}$ will be stable on all levels of observed $\mathbf{W}$ and $\mathbf{Z}$. In the cat and dog example, the environment (indoors vs. outdoors) would be a single variable $Z$ (from the set

of spurious variables $\mathbf{Z}$). To make the predictor only learn the stable effect from target $Y$ (cat, dog) to the image $X$, we would require our model to satisfy the sufficient condition $\hat{Y} \perp Z \mid Y$.

Our second contribution introduces DISCO, a new approach to achieve the conditional independence $\hat{Y} \perp \mathbf{W}, \mathbf{Z} \mid Y$ through regularization during training, see Sec. 3. DISCO is based on the distance correlation (Székely et al., 2007; Székely & Rizzo, 2014) and the conditional distance correlation (Wang et al., 2015) metrics. We demonstrate that DISCO is competitive among recent state-of-the-art techniques across a chosen set of regression experiments. Our code is available at https://github.com/anonresearch123456/DISCO.

## 2 CAUSAL INVARIANCE ON NON-DIRECT PATHS

First, we introduce the required terminology and theory. We further extend the standard fairness model (Zhang & Bareinboim, 2018; Plecko & Bareinboim, 2022) and allow any spurious correlation connecting $Y$ and $X$ in the anti-causal prediction setting, thus creating the Standard Anti-Causal Model (SAM). Then, we show that a predictor, which satisfies the conditional independence $\hat{Y} \perp \mathbf{W}, \mathbf{Z} \mid Y$, will only have a direct effect from $Y$ to $\hat{Y}$, while the counterfactual indirect and spurious effects will be zero, and the direct effect will be counterfactually constant on all observed $\mathbf{Z}$ and $\mathbf{W}$.

**Structural Causal Model (SCM).** SCMs are quadruples $(\mathbf{V}, \mathcal{F}, \mathbf{U}, P_{\mathbf{U}})$ where $\mathbf{U}$ are the exogenous variables. Their distribution $P_{\mathbf{U}}$ is defined outside of the model. $\mathbf{V}$ are the random variables in the model. They are composed of other variables from $\mathbf{V}$ and get their uncertainty "injected" by exogenous variables $\mathbf{U}$. $\mathcal{F} = \{f_1, ..., f_n\}$ are functions such that $v_i \leftarrow f_i(pa(x_i), u_i)$, where $pa(v_i) \subseteq \mathbf{V}$ are the parents of $v_i$, and $u_i \subseteq \mathbf{U}$ are the exogenous variables influencing $v_i$.

We assume a one-to-one correspondence between nodes in a causal graph and random variables. We, therefore, use capital letters, such as $X$ for random variables, and lowercase $x$ to indicate values that $X$ can attain in its respective range. If not specified otherwise, bold letters $\mathbf{X}$ will denote random vectors (or sets of nodes in a graph), and their values are denoted with $\mathbf{x}$.

Interventional distributions, induced by $do(X = x)$, will be denoted by subscripts, e.g., $P(Y_x)$. Unit-level counterfactuals will be explicitly denoted by $Y_x(\mathbf{u})$ given realizations of all exogenous variables $\mathbf{u}$; this expression is related to potential outcomes. Counterfactual expressions given a set of observations will be denoted by $P(Y_x \mid X = x')$. While this latter expression is probabilistic (conditioned over a population), the expression $Y_x(\mathbf{u})$ is deterministic (given $\mathbf{u}$). As these notions can be easily confused, we refer the reader to Bareinboim et al. (2022); Plecko & Bareinboim (2022). For simplicity, we use summation for discrete values in our proofs. However, summation can easily be replaced by integration and the probability measure with the density when dealing with real-valued random variables/vectors, assuming the density functions always exist.

**Standard Anti-Causal Model (SAM)** Fig. 1a shows the graph for the Standard Anti-Causal Model, which is a generic causal graph for many anti-causal prediction tasks. $X$ is the input random variable for a prediction task, $Y$ is the outcome, and $\hat{Y}$ is the prediction made by a fixed model. $\mathbf{Z}$ is a group of nodes that lie on alternative open paths between $Y$ and $X$ and are non-causal, and $\mathbf{Z}$ is a valid adjustment set for $(Y, X)$ A.1. We denote this by using $Y - Z - X$ and point to appendix A for a more detailed view. We especially allow any connections between elements in $\mathbf{Z}$ as long as it stays a valid adjustment set. $\mathbf{W}$ are (intermediate or indirect) variables that lie on causal-paths between $Y$ and $X$. For the sake of simplicity, we assume there are no arrows between elements of $\mathbf{W}$. Furthermore, we assume that our task's only stable relevant information flows directly from $Y$ to $X$, rendering the nodes in $\mathbf{W}, \mathbf{Z}$ distracting/undesired. Relaxation and extension of these assumptions will be discussed in Sec.B.4.

If we trained a model oblivious to the SAM using classical empirical risk minimization (ERM), the predictor would capture all information to make the best possible prediction. This would especially mean that our model would rely on all paths leading from $Y$ to $X$, thereby learning biased effects irrelevant to the task. Since we do not want to generatively model the entire prediction task, we focus on the effect of $Y$ on the predictions $\hat{Y}$ of a model and define the path $Y \to X \to \hat{Y}$ as the direct path.

We can measure how much $\hat{Y}$ changes when different values of $Y$ are observed. Translating this to a measure, we can get the total variation (TV) which reads $TV_{y_0,y_1}(\hat{y}) = P(\hat{y} \mid y_1) - P(\hat{y} \mid y_0)$. While we want to maximize the TV with ERM for different observations $y_0 \neq y_1$, we also want to ensure that only relevant information causes the large difference. Plecko & Bareinboim (2022) recently showed a useful decomposition of this metric into counterfactual direct (ctf-DE), spurious (ctf-SE) and indirect (ctf-IE) effects.

**Counterfactual-(TE,DE,IE,SE) (Plecko & Bareinboim, 2022):** The counterfactual total effect (*ctf-TE*) measures the difference of probabilities of $\hat{Y}$ when doing atomic interventions $y_0$ and $y_1$ on $Y$, having observed $Y = y$ before. The expression reads

$$ctf\text{-}TE_{y_0,y_1}(\hat{y} \mid y) = P(\hat{y}_{y_1} \mid y) - P(\hat{y}_{y_0} \mid y). \tag{1}$$

It captures all counterfactual effects given $y$ that lead from $Y$ to $\hat{Y}$. In SAM, *ctf-TE* also includes effects over $\mathbf{W}$. Note again that we do not perceive $X$ as an intermediate by definition, as it is our input.

To separate the counterfactual direct effects (*ctf-DE*) from the indirect ones (*ctf-IE*), we can further write

$$ctf\text{-}DE_{y_0,y_1}(\hat{y} \mid y) = P(\hat{y}_{y_1,w_{y_0}} \mid y) - P(\hat{y}_{y_0} \mid y) \tag{2}$$

$$ctf\text{-}IE_{y_1,y_0}(\hat{y} \mid y) = P(\hat{y}_{y_1,w_{y_0}} \mid y) - P(\hat{y}_{y_1} \mid y). \tag{3}$$

Finally, there is a non-causal part complementing all effects of $Y$ on $\hat{Y}$. This is called the counterfactual spurious effect (*ctf-SE*) and is expressed as

$$ctf\text{-}SE_{y_1,y_0}(\hat{y}) = P(\hat{y}_{y_1} \mid y_1) - P(\hat{y}_{y_1} \mid y_0), \tag{4}$$

capturing all non-directed effects that are present between $Y$ and $\hat{Y}$.

**Proposition 1.** *The TV can be decomposed (Plecko & Bareinboim, 2022) into direct, indirect, and spurious components by*

$$TV_{y_0,y_1}(\hat{y}) = ctf\text{-}DE_{y_0,y_1}(\hat{y} \mid y) - ctf\text{-}IE_{y_1,y_0}(\hat{y} \mid y) - ctf\text{-}SE_{y_1,y_0}(\hat{y}). \tag{5}$$

*The proof is in appendix B.1.*

This insightful decomposition shows that a classical empirical risk-minimizing predictor would blindly maximize the TV of $Y$ on $\hat{Y}$. We see that such a predictor will use all information available, including indirect and spurious ones. The following two propositions show that a certain conditional independence criterion is sufficient for a predictor to have no *ctf-IE* and *ctf-SE*. At the same time, the *ctf-DE* stays constant for any observations of $\mathbf{w}$ and $\mathbf{z}$ if the condition is satisfied.

**Proposition 2.** *SAM is given. Assume we have a prediction model that maps from the input $X$ to our target. Let's treat the predictions of our model as a random variable called $\hat{Y}$. If the prediction satisfies $\hat{Y} \perp \mathbf{W}, \mathbf{Z} \mid Y$, then $ctf\text{-}IE_{y_1,y_0}(\hat{y} \mid y) = ctf\text{-}SE_{y_1,y_0}(\hat{y}) = 0$, for any $y, y_0, y_1, \hat{y}$ in their ranges. The conditional independence criterion even ensures a stricter criterion, namely, $\hat{Y} \perp \mathbf{W}, \mathbf{Z} \mid Y \implies ctf\text{-}IE_{y_1,y_0}(\hat{y} \mid y, \mathbf{z}, \mathbf{w}) = 0$, for any $y, y_0, y_1, \hat{y}, \mathbf{w}, \mathbf{z}$ in the ranges of their respective random variables. The proof is in appendix B.2*

**Proposition 3.** *SAM is given. Assume we have a predictor and its output represented as a random variable $\hat{Y}$, such that $\hat{Y} \perp \mathbf{W}, \mathbf{Z} \mid Y$ holds. This predictor will have a constant direct effect for fixed $y_0, y_1$, for any $\mathbf{w}, \mathbf{z}$. This is compactly expressed as $ctf\text{-}DE_{y_0,y_1}(\hat{y} \mid y, \mathbf{w}, \mathbf{z}) = ctf\text{-}DE_{y_0,y_1}(\hat{y} \mid y, \mathbf{w}', \mathbf{z}')$, for any $y_0, y_1, \mathbf{w}, \mathbf{z}, \mathbf{w}', \mathbf{z}'$ in the respective ranges. The proof is in appendix B.3.*

Thus, to obtain a stable predictor that relies solely on the direct effects from $Y$ on $X$, we can perform a constrained optimization of the following form:

$$\begin{aligned} \min_{\theta} \quad & \mathbb{E}_{(X,Y)} \left[ L\left(Y, g_\theta(X)\right) \right] \\ \text{s.t.} \quad & \hat{Y} \perp \mathbf{W}, \mathbf{Z} \mid Y, \end{aligned} \tag{6}$$

where $\hat{Y} = g_\theta(X)$, $g$ is a learning model parametrized by $\theta$ and $L$ is a classical loss function (mean squared error, cross-entropy, etc.). In the appendix in B.4, we show how we can change/relax the assumptions made in SAM and what consequences are implied by different changes.

## 3 DISCO: CONDITIONAL INDEPENDENCE WITH DISTANCE CORRELATION

In the previous section, we showed that a predictor $g_\theta$ utilizes only the direct path of information from $Y$ to $X$ if we can ensure the conditional independence: $\hat{Y} \perp \mathbf{W}, \mathbf{Z} \mid Y$. Directly solving Eq.(6) is generally impractical, as finding closed-form solutions is not trivial and generally not possible for black-box methods such as neural networks. Thus we relax the problem into a regularized problem and optimize the model with $\min_\theta \; \mathbb{E}_{(X,Y)} \left[ L\left(Y, g_\theta(X)\right)\right] + \lambda Reg(\hat{Y}, \mathbf{Z}, \mathbf{W}, Y)$. We refer to the variables with undesired effects $\mathbf{Z}, \mathbf{W}$ as bias variables in the following for simplification.

**Related Work on Conditional Independence Regularization**   Recent research increasingly focuses on achieving conditional independence by a regularization penalty for neural networks. Maximum Mean Discrepancy (MMD) and its conditional variants (c-MMD) are used by Makar et al. (2022); Veitch et al. (2021); Makar & D'Amour (2022); Kaur et al. (2022) and are shown to be effective in handling binary and categorical attributes in classification settings. However, these methods exhibit inefficiencies as they necessitate dividing batches into sub-samples (or strata) according to all combinations of label values (e.g., $y_0, y_1$) and bias attributes (e.g., $a_1, a_2, a_3$). This approach is particularly cumbersome for scenarios with multi-bias, multi-class, or biases of many categories. In contrast, methods built on the Hilbert-Schmidt-Criterion (HSIC) Gretton et al. (2005; 2007), and its conditional variant Fukumizu et al. (2007); Zhang et al. (2012); Park & Muandet (2020) (HSCIC) were proposed to achieve the conditional independence criterion in regression and classification tasks. Zheng & Makar (2022) extended HSIC to classification tasks with multiple, mixed-valued biases, and Quinzan et al. (2022) adapted HSCIC with conditional independence regularization for regression tasks, relying on multiple ridge regressions. Lastly, Pogodin et al. (2022) proposed CIRCE, a modified variant estimating a notion similar to HSCIC. The method reduces the number of in-batch regressions via the estimation of hyperparameters on a split of the training set.

**Current Challenges**   We conclude that most causally inspired deep learning methods were proposed for classification. To this end, we focus on regression and observe that existing methods like HSCIC and CIRCE introduce a significant number of additional hyperparameters: A regularization strength $\lambda$, three bandwidth parameters for the (Gaussian) kernels, and a ridge regression parameter.

**Our Contribution**   To overcome the limitations of current conditional independence regularization techniques, we propose to regularize the training of deep learning models with Conditional Distance Correlation ($cdCor$) Wang et al. (2015); Póczos & Schneider (2012), which is a measure of conditional dependence in arbitrary dimensions. The general idea is to bring the $cdCor$ as close to zero as possible since $cdCor(X, Y; Z) = 0 \iff X \perp Y \mid Z$. $cdCor$ is a relatively uncommon measure, primarily employed in biomedical settings for feature screening and related applications (Wang et al., 2015; Song et al., 2020). To our knowledge, this is the first instance of its use in a deep learning context. In the following subsections, we derive a method for estimating it batch-wise to suit our purposes.

**Notation and Definitions**   We deal with random vectors $X \in \mathbb{R}^p$, $Y \in \mathbb{R}^q$, and $Z \in \mathbb{R}^r$. If applicable, Z will be the random vector we will condition on. Lowercase $z$ denotes the values the random vector $Z$ can take in its range. For each random vector, we assume the existence of probability densities, that each random vector has finite moments (e.g., $E[X] < \infty$), and that $X$ and $Y$ have finite conditional moments given $Z$ (e.g., $E[X \mid Z = z] < \infty$, for any $z$ in the range of $Z$). We also consider sets of random samples denoted as $(\mathbf{X}, \mathbf{Y}, \mathbf{Z}) = \{(X_k, Y_k, Z_k)\}_{k=1}^n$, where we assume they are iid concerning the random vectors $X$, $Y$, and $Z$, respectively.

The joint characteristic function is $\phi_{X,Y}(t, s) = \mathbb{E}\left[e^{i\langle t, X\rangle + i\langle s, Y\rangle}\right]$, where $t$ and $s$ are in the same space as the inputs $X$ and $Y$. The marginal characteristic functions can be written as $\phi_{X,Y}(0, s)$ and $\phi_{X,Y}(t, 0)$, respectively. The joint conditional characteristic function will read $\phi_{X,Y|Z=z}(t, s) = \mathbb{E}\left[e^{i\langle t, X\rangle + i\langle s, Y\rangle} \mid Z = z\right]$, and the marginals read $\phi_{X,Y|Z=z}(0, s)$ and $\phi_{X,Y|Z=z}(t, 0)$. If not specified otherwise, $||\cdot||$ will denote the Euclidean norm. Note that this section's proofs for the lemmata can be found in Székely et al. (2007); Wang et al. (2015).

## 3.1 DISTANCE CORRELATION FOR CONDITIONAL INDEPENDENCE

Conditional distance correlation makes use of the notion that two random vectors $X$ and $Y$ are conditionally independent of $Z$ if and only if their joint characteristic functions conditioned on $Z$ can be decomposed into the products of their marginals conditioned on $Z$: $\phi_{X,Y|Z=z}(t,s) = \phi_{X,Y|Z=z}(0,s)\phi_{X,Y|Z=z}(t,0)$ for all $z$. The idea is to find a measure that quantifies how far the product of the marginals deviates from the true underlying joint characteristic function.

**Measuring Conditional Dependence between Random Vectors:** We define a measure $\mathcal{V}$ of dependence between random vectors $X$ and $Y$ given $Z$, $\mathcal{V}$, using the $||\cdot||_w$-norm (in the weighted $L_2$ space of complex functions on $\mathbb{R}^{p+q}$), such that

$$\mathcal{V}_w^2(X,Y \mid Z=z) = \int_{\mathbb{R}^{p+q}} \left|\phi_{X,Y|Z=z}(t,s) - \phi_{X|Z=z}(t)\phi_{Y|Z=z}(s)\right|^2 w(t,s)\,dt\,ds, \quad (7)$$

$$\mathcal{V}_w^2(X,Y \mid Z) = \int p(z)V(X,Y \mid Z=z)\,dz, \quad (8)$$

where $w(t,s)$ is a weight function, $|\cdot|$ is the (complex) magnitude, and $p(z)$ is the marginal density function of $Z$. Note the difference between $\mathcal{V}_w^2(X,Y \mid Z=z)$, the "local" measure of conditional dependence, and $\mathcal{V}_w^2(X,Y \mid Z)$, the "global" measure of conditional dependence. We find the latter expression useful, giving us a single number measuring the deviation from the conditional independence criterion. Given a suitable $w$, $\mathcal{V}^2$ equals zero when the joint distribution perfectly decomposes into the product of its marginals - in other words, we have conditional independence.

Although irrelevant to our practical discussion and utilization of the conditional distance correlation metric, we want to comment on the choice of $w$. Feuerverger (1993); Székely et al. (2007) argue about the suiting choice of weight functions $w$ and derive a very specific one. Their choice, which has an impractically complex form for our presentation, has the nice property that the integral will not evaluate to zero in trivial cases. Additionally, the specific $w$ leads to a very simple form for the unconditional distance correlation; the conditional distance correlation partially benefits from this property, and we discuss the sample estimates in subsection 3.2. From here, we assume $w$ as chosen as in Székely et al. (2007) and drop the subscript in $\mathcal{V}^2(X,Y \mid Z)$.

**Conditional Distance Covariance and Distance Correlation** Given random vectors $X, Y$, and $t, s$ in the same space, the measure of conditional dependence given $Z$ will be called conditional distance covariance $\mathcal{V}^2(X,Y \mid Z)$. The corresponding conditional distance correlation measure will be defined as

$$R(X,Y \mid Z=z) = \sqrt{\frac{\mathcal{V}^2(X,Y \mid Z=z)}{\mathcal{V}^2(X,X \mid Z=z) \cdot \mathcal{V}^2(Y,Y \mid Z=z)}}. \quad (9)$$

We can define the same for the global conditional distance correlation measure as $R(X,Y \mid Z) = \int p(z)R(X,Y \mid Z=z)$. The first advantage of the choice of our dependence measures $R(X,Y \mid Z)$ is given in the following lemma (see Appendix C.1 for the proof).

**Lemma 1.** *Given random vectors $X,Y,Z$ we have 1) $0 \leq R(X,Y \mid Z=z) \leq 1$ for any $z$; 2) $R(X,Y \mid Z=z) = 0$ for any $z \iff X \perp Y \mid Z$; 3) $R(X,Y \mid Z=z)$ is invariant in orthogonal transformations of $X, Y$, and especially scale invariant.*

Property 2) in Lemma 1 is the most useful one as it directly translates to our regularization goal. The only remaining challenge is to estimate the conditional distance correlation from a given sample.

## 3.2 SAMPLE ESTIMATION

The unconditional distance correlation has a very simple sample estimate due to the choice of $w$ (Feuerverger, 1993; Székely et al., 2007). The conditional variant, however, does not allow for a simple closed-form expression. We, therefore, propose using the Nadaraya-Watson estimation (kernel regression) Wasserman (2006); Ullah & Pagan (1999); Ruppert & Wand (1994) for the conditional distance correlation. The following representation makes it clear that this estimation is applicable.

**Lemma 2.** *Given random vectors $X, Y, Z$ and independent copies $(X', Y', Z')$ and $(X'', Y'', Z'')$, the conditional distance covariance can be expressed as:* $V^2(X, Y \mid Z = z) = S_1^z + S_2^z - 2S_3^z$ *where:*

$$S_1^z = E_{X,Y}\left[E_{X',Y'}\left[||X - X'||\,||Y - Y'|| \mid Z' = z\right], Z = z\right],$$

$$S_2^z = E_{X,X'}\left[||X - X'|| \mid Z = z, Z = z'\right] + E_{Y,Y'}\left[||Y - Y'|| \mid Z = z, Z' = z\right],$$

$$S_3^z = E_{X,Y}\left[E_{X'}\left[||X - X'|| \mid Z' = z\right]\,E_{Y''}\left[||Y - Y''|| \mid Z'' = z\right] \mid Z = z\right].$$

This expression allows us to directly apply the Nadaraya-Watson estimator. We aim to estimate the local conditional distance correlation $cdCor_z$ and thereby derive the global metric $cdCor$. For simplicity, assume $Z$ is one-dimensional (all of the following is easily transferred to the multi-dimensional case). We define $k_i^h(z) = K_h(z - Z_i) / \sum_i K_h(z - Z_i)$, where $K_h(t) = \frac{1}{h}K(\frac{t}{h})$, $K$ is the Gaussian kernel function with bandwidth $h$, which is a non-negative, real-valued function that integrates to one and is symmetric around zero, and $Z_i$ is the i-th data point of the set $\mathbf{Z}$, $z$ an arbitrary evaluation point in the valid range. We use the plug-in estimate for $S_1^z, S_2^z$, and, $S_3^z$ to arrive at the equations

$$\hat{S}_1^z = \sum_i \sum_j k_i^h(z)k_j^h(z)||X_i - X_j||\,||Y_i - Y_j|| \tag{10}$$

$$\hat{S}_2^z = \sum_i \sum_j k_i^h(z)k_j^h(z)||X_i - X_j|| \sum_i \sum_j k_i^h(z)k_j^h(z)||Y_i - Y_j|| \tag{11}$$

$$\hat{S}_3^z = \sum_i \sum_j \sum_l k_i^h(z)k_j^h(z)k_l^h(z)||X_i - X_j||\,||Y_i - Y_l||. \tag{12}$$

**Definition 1.** *We define the local estimator $cdCor_z(\mathbf{X}, \mathbf{Y}) = \hat{S}_1^z + \hat{S}_2^z - 2\hat{S}_3^z$. We observe that the global expression is $\mathcal{V}_w^2(X, Y \mid Z) = E_Z[\mathcal{V}_w^2(X, Y \mid Z = z)]$, and we use the law of large numbers to arrive at the final global estimator $cdCor(\mathbf{X}, \mathbf{Y}; \mathbf{Z}) = \frac{1}{n}\sum_{i=1}^n cdCor_{z_i}(\mathbf{X}, \mathbf{Y})$.*

Bias, variance, and asymptotic behavior of Nadaraya-Watson estimators are well-studied. The interested reader is referred to Wasserman (2006); Ullah & Pagan (1999); Ruppert & Wand (1994) for analyses, including the choice of bandwidth and assumptions on derivatives of the expectations considered. From a different angle, Wang et al. (2015) define a computationally more involved V-statistic which is equivalent to the estimator $cdCor_z(\mathbf{X}, \mathbf{Y})$, and as a consequence derive $cdCor_z(\mathbf{X}, \mathbf{Y}) \xrightarrow[n \to \infty]{\text{P}} V^2(X, Y \mid Z = z)$.

Coming back to our initial goal from this section, we arrive at the optimization procedure $\min_{\theta_g} \mathbb{E}_{(X,Y)}\left[L(Y, g_\theta(X))\right] + \lambda \cdot cdCor(\hat{Y}, \mathbf{Z} \cup \mathbf{W}; Y)$, with the variables as defined in the SAM. We note that we estimate the loss on a batch level as we work with deep neural networks and try to keep the batch size as large as possible, given the dataset.

# 4 EXPERIMENTS AND RESULTS

We evaluate the efficacy of DISCO in tackling the challenge of providing invariance in anti-causal settings that SAM captures. We selected two challenging 2D image tasks from Pogodin et al. (2022) on the Extended YaleB Georghiades et al. (2001); Yale (2001) and dSprites Matthey et al. (2017) datasets. Further, we use our challenging 3D image/volume simulation dataset incorporating a vector-valued unwanted indirect effect that we try to mitigate. We compare with CIRCE Pogodin et al. (2022) and HSCIC Quinzan et al. (2022), which are designed to regularize for conditional independence in regression settings.

We want to emphasize that many other bias mitigation methods (Makar et al., 2022; Kaur et al., 2022) or domain generalization algortihms Sagawa et al. (2019); Arjovsky et al. (2019) exist, but most of them are made for classification tasks or solely allow to model categorical bias variables. Our case, however, is well-suited for the often neglected task of a continuous label ($Y$) and vector-valued bias variables (continous, categorical, mixed). Thus, the benchmark methods we chose follow the setup of Pogodin et al. (2022) which exactly treats these cases.

The experimental procedure for all datasets is as follows: First, we train all models on the biased domain (in-domain) with a hyperparameter search. All methods (CIRCE, HSCIC, DISCO) use the

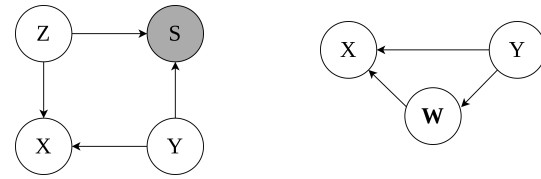

(a) Setup for dSprites and YaleB    (b) Setup for 3D simulation

Figure 2: (a) Causal setup for dSprites and YaleB. The shaded node S denotes that we implicitly condition on it, resulting in a selection bias. X: 2D image, Y: target, Z: spurious variable. (b) Causal setup for the 3D simulation. X: 3D images, **W**: set of unwanted intermediate effects, Y: label.

exact same neural network backbone. Second, for each regularization strength $\lambda$ given a model, we choose the $\lambda$ with the best in-domain validation loss. We report the test mean squared errors (MSE) for each $\lambda$, following the procedure described in Pogodin et al. (2022), where the test domain will be unbiased (out-of-domain, or OOD). The red-dotted line in Figs 3,4,5 is the performance of a baseline (backbone network without regularization) trained on an OOD dataset. We detail why this strategy is valid and sensible in the appendix D.1. The details to the backbone for each dataset, and all the hyperparameters for each experiment can be found in the appendix D.

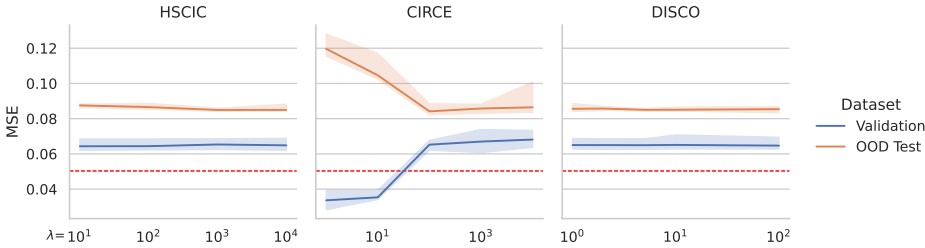

Figure 3: Median MSE for each $\lambda$ on YaleB. Errors show min and max across 5 random seeds.

**Extended YaleB**   This dataset comprises 16,128 images of 28 human subjects under 9 poses and 64 illumination conditions Yale (2001). It directly extends the original YaleB dataset Georghiades et al. (2001) with 18 new subjects. We follow the exact experimental setup of Pogodin et al. (2022), define the poses as our target $Y$, and treat them as continuous variables. The distractor $Z$ will be the illumination angles that have direct effects on the image $X$. $U$ denotes exogenous noises we cannot control and is left out of the graph in Fig. 2a. $S$ is a sampling bias variable that non-linearly correlates $Y$ and $Z$. The equations are

$$S = \begin{cases} 1, & \text{if } \left| Z - 0.5(Y - \epsilon Y^2) \right| < 1 \\ 0, & \text{otherwise} \end{cases} \quad \begin{aligned} Y &= f_Y(U_Y), & Z &= f_Z(U_Z) \\ X &= f_X(Y, Z, U_X) & \epsilon &= \mathcal{U}\{-1, 1\}. \end{aligned}$$

The in-domain and OOD-domain datasets follow the distributions $P(X, Y, Z \mid S = 1)$ and $P(X, Y, Z)$, respectively. The OOD-domain has no dependence between $Z$ and $Y$.

The results in Fig. 3 show how DISCO performs equally well as the other two methods. We observe no significant differences between the models, particularly at the optimal setting for $\lambda$, leading us to conclude that DISCO is a viable competitor in this experiment. We especially note that HSCIC and CIRCE follow the same trend as in the work of Pogodin et al. (2022).

**dSprites**   dSprites is a simulation dataset for 2D images that contain shapes of different sizes, colors, shapes, x- and y-positions, and a few more attributes Matthey et al. (2017). Again, we follow the setup of Pogodin et al. (2022) and introduce the following structural equations, where the in-domain and OOD-domain distributions are $P(X, Y, Z \mid S = 1)$ and $P(X, Y, Z)$, respectively. $Y$ is the y-coordinate of an object, the distractor $Z$ is the x-coordinate (see Fig. 2a for the causal graph). Note

that we leave out the equations for $X, Y$, and $Z$ below as they are syntactically the same (although their noises $U$ differ)

$$S = \begin{cases} 1, & \text{if } \left| Z - \epsilon^2 - Y \right| < 1 \\ 0, & \text{otherwise} \end{cases} \qquad \epsilon = \begin{cases} -\rho, \text{with p=0.5} \\ \rho, \text{with p=0.5} \end{cases} \qquad \rho = \mathcal{U}\{0, 3\}.$$

Fig. 4 demonstrates that all methods achieve competitive comparable results for the best hyperparameter configuration. For HSCIC and DISCO, the trend along choices of $\lambda$ indicates that hyperparameters for lower ranges might be more beneficial, as both validation and OOD test MSE decline. For CIRCE and DISCO we see the potential for lower values which indicates better invariances, while the overall median is is comparable among all methods. We want to note the reader that the results for HSCIC have the same trend as in Pogodin et al. (2022) but the trend for CIRCE seems quite different from what they report. We have no obvious explanations for this apart from differences in number of maximal epochs. We used a smaller number of epochs due to time limitations but we had a close look on the development of losses and chose appropriate thresholds where we could not see any further improvements. Since Pogodin et al. (2022) used early stopping, we believe they arrived at similar actual epoch runs.

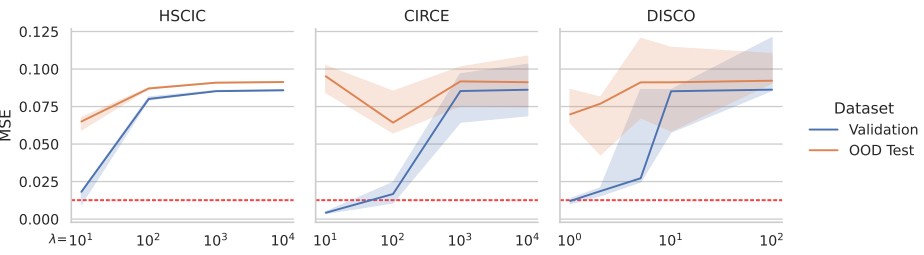

Figure 4: Median MSE for each $\lambda$ on dSprites. Errors show min and max across 5 random seeds.

**3D simulation** Finally, we use a simulation dataset, where we create ellipsoids on a $32^3$ grid. The ellipsoids have a fixed x:y:z ratio of 1:2.5:0.5 for the axes in three dimensions. The target $Y$ is the rotation angle in this experiment (we use the same value for all three rotation axes), the unwanted mediators are $\mathbf{W}$ consisting of the x-,y- and z-positions of the ellipsoid centers, the brightness of the ellipsoid and a single scale factor for the ellipsoid size in all dimensions. The structural equations are stated in the appendix D.4. In this experiment, the causal graph in Fig. 2b differs from the previous ones, and the mediator is higher dimensional. For the OOD domain, we interventionally fix the mediators $\mathbf{W}$ such that the intermediate effects from the biases are removed, see D.4.

Fig. 5 shows that DISCO performs slightly better than HSCIC, while CIRCE generally struggles to achieve invariance. We believe the challenge of this problem comes from the fact that the mediating bias variables have strong cues in the image, and the bias variable is vector-valued in this case of dimension five.

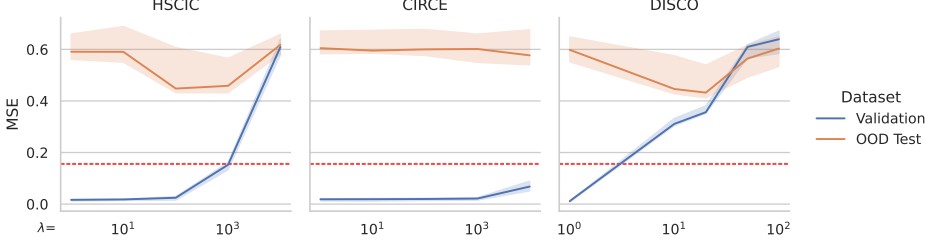

Figure 5: Median MSE for each $\lambda$ on 3D simulation. Errors show min and max across 5 rnd. seeds.

## 5 DISCUSSION

We proposed a unified description of the anti-causal prediction setting called SAM. Utilizing this causal analysis model, we derived that the conditional independence criterion $\hat{Y} \perp \mathbf{W}, \mathbf{Z} \mid Y$ is sufficient to provide counterfactual invariance in anti-causal scenarios. In our case, we concluded that if one wanted to remove all effects from $\mathbf{W}$ and $\mathbf{Z}$, there is no need to distinguish between these variables conceptually.

At a first glance, distinguishing between variables $\mathbf{W}$ and $\mathbf{Z}$ might seem unnecessary as they are treated equally in the conditional independence term. However, if we look at the causal literature, we clearly see that this kind of pathway analysis is a useful and well-studied tool (Pearl, 2022; Shpitser & VanderWeele, 2011; Plecko & Bareinboim, 2022). It is especially natural to separate the directionalities of causal flows: the flow of information can be direct, mediated, or entirely spurious. The nodes in $\mathbf{W}$ have a completely different meaning than the ones in $\mathbf{Z}$. We, therefore, see the result of the simple conditional independence as something particularly handy, as variables in both sets can be treated equally under certain assumptions. Lastly, although we kept our experimental setups straightforward, one can create situations where mediated effects in $\mathbf{W}$ can be desirable. How to proceed from there is discussed in the appendix B.4 where we assume we want to retain some of the mediated effects.

We believe this causal framework is useful for understanding different setups in bias mitigation, domain generalization, and similar. This obviously requires making causal assumptions, but we see it as a chance to be precise and more scientific rather than perceiving it as a burden. Future efforts can be put into how one can actually quantify and estimate the causal effects on the decisions using pathway analysis. One can, for example, theoretically estimate the causal effects of different variables on the final decision, leading to useful measures of invariance or dependence on undesired paths.

Moreover, we proposed optimizing the conditional independence criterion by utilizing the conditional distance correlation metric. Given our experimental results, we conclude that DISCO is competitive and slightly better than the competing models on certain datasets. DISCO, however, also has its downsides that need to be addressed. While HSCIC and CIRCE perform kernel (ridge) regressions that cost $O(b^2)$ given a batch size $b$, DISCO suffers from overhead as it has a compute and memory requirement of $O(b^3)$. The difference is noticeable depending on the batch sizes one operates on, and we accept this as a drawback of our method.

While DISCO incurs a higher computational cost of $O(b^3)$ compared to the $O(b^2)$ of HSCIC and CIRCE for a batch size $b$, it offers a significant advantage in terms of hyperparameter optimization. DISCO requires tuning only a single parameter for its Nadaraya-Watson estimation, whereas HSCIC and CIRCE involve three such hyperparameters. This reduction in the hyperparameter search space partially mitigates the computational overhead of DISCO. Furthermore, as demonstrated in Fig. D.7, competing models can be highly sensitive to these parameters. Consequently, DISCO's simpler hyperparameter landscape potentially leads to more robust model selection procedures and other training protocols. While the computational complexity trade-off ($O(b^3)$ vs. $O(b^2)$) can be substantial for large batch sizes, the reduction in hyperparameter search iterations (from $p^3$ to $p$ for $p$ parameter values) offers a notable efficiency compensation in the overall experimental pipeline.

Overall, we conclude that SAM can readily describe different anti-causal prediction scenarios for bias mitigation and enable the derivation of the simple conditional independence criterion. At the same time, DISCO is a competitive alternative to the established kernel-based methods for invariance tasks.

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

## A  CAUSAL GRAPH AND ASSUMPTIONS

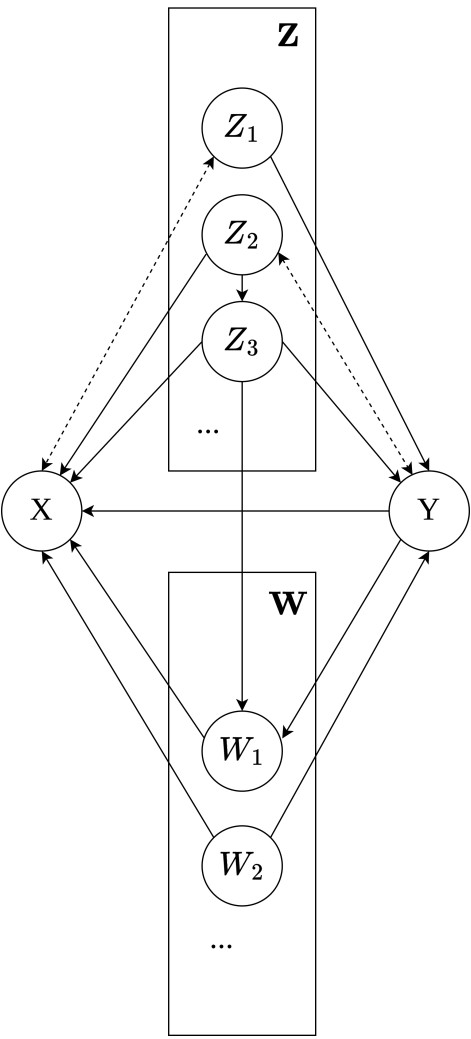

Figure A.1: The main setting we consider in anti-causal prediction. Y is the target we want to predict using X only. **W** is a set of variables that only permits mediated effects from Y to X. **Z** contains nodes that lie on open paths between Y and X but are non-causal. Nodes in **Z** are allowed to have arbitrary connections in between them, as long as **Z** stays a valid adjustment set for (Y,X), and the graph is still a directed acyclic graph (DAG). Nodes in **W** are mediators and are not allowed any other connections. We discuss the relaxation of the latter condition in Sec. B.4. The meaning of dashed bi-directed arrows is explained in A.2.

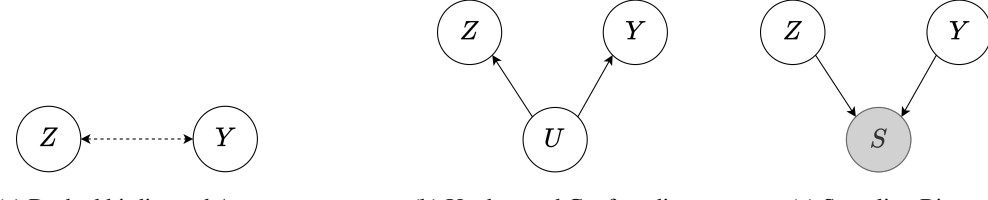

(a) Dashed bi-directed Arrow      (b) Unobserved Confounding      (c) Sampling Bias

Figure A.2: We detail the meaning of dashed bi-directed arrows as in (a). Figure (b) shows the regular meaning of dashed bi-directed arrows used in many texts. We extend this notation here and also allow the sampling bias in (c), where the entire observed distribution is implicitly conditioned on (marked with shading) the variable $S$, a collider, which opens a new undirected path between $Z$ and $Y$.

## A.1    VALID ADJUSTMENT SET

**Characterization of Valid Adjustment Sets Shpitser et al. (2012)** Let $Z$ be a set of nodes in a causal graph. Then, $Z$ is a valid adjustment set for $(Y, X)$ (implying $Y \cap X = \text{pa}_Y \cap X = \emptyset$) if and only if it satisfies the following conditions:

   (i) $Z$ contains no node $R \notin Y$ on a proper causal path from $Y$ to $X$ nor any of its descendants in $G_Y$,

   (ii) $Z$ blocks all non-directed paths from $Y$ to $X$.

## B    CAUSAL INVARIANCE AS CONDITIONAL INDEPENDENCE

In the following, we will prove the propositions in Sec.2.

### B.1    TOTAL VARIANCE DECOMPOSITION

**Proposition 1 (TV-Decomposition)** The TV can be decomposed into direct, indirect, and spurious components by

$$TV_{y_0,y_1}(\hat{y}) = ctf\text{-}DE_{y_0,y_1}(\hat{y}|y) - ctf\text{-}IE_{y_1,y_0}(\hat{y}|y) - ctf\text{-}SE_{y_1,y_0}(\hat{y}). \tag{13}$$

*Proof.* We first show that $ctf\text{-}TE_{y_0,y_1}(\hat{y}|y_0)$ can be split into its direct and indirect components as given in definition 2.

$$ctf\text{-}TE_{y_0,y_1}(\hat{y}|y_0) = P(\hat{y}_{y_1}|y_0) - P(\hat{y}_{y_0}|y_0) \tag{14}$$
$$= P(\hat{y}_{y_1}|y_0) - P(\hat{y}_{y_1,w_{y_0}}|y_0) + P(\hat{y}_{y_1,w_{y_0}}|y_0) - P(\hat{y}_{y_0}|y_0) \tag{15}$$
$$= ctf\text{-}DE_{y_0,y_1}(\hat{y}|y_0) - ctf\text{-}IE_{y_1,y_0}(\hat{y}|y_0). \tag{16}$$

We now can finally show that the total variance can be split into total effects and spurious effects, as given in the following:

$$TV_{y_0,y_1}(\hat{y}) = P(\hat{y}|y_1) - P(\hat{y}|y_0) \tag{17}$$
$$= P(\hat{y}|y_1) - P(\hat{y}_{y_1}|y_0) + P(\hat{y}_{y_1}|y_0) - P(\hat{y}|y_0) \tag{18}$$
$$= P(\hat{y}_{y_1}|y_0) - P(\hat{y}_{y_0}|y_0) + P(\hat{y}_{y_1}|y_1) - P(\hat{y}_{y_1}|y_0) \tag{19}$$
$$= ctf\text{-}TE_{y_0,y_1}(\hat{y}|y_0) - ctf\text{-}SE_{y_1,y_0}(\hat{y}). \tag{20}$$

In total, we arrive at

$$TV_{y_0,y_1}(\hat{y}) = ctf\text{-}DE_{y_0,y_1}(\hat{y}|y_0) - ctf\text{-}IE_{y_1,y_0}(\hat{y}|y_0) - ctf\text{-}SE_{y_1,y_0}(\hat{y}). \tag{21}$$

$\square$

**Proposition 4.** *($ctf\text{-}IE_{y_1,y_0}(\hat{y}|y_0,\mathbf{w},\mathbf{z})$ is stronger than $ctf\text{-}IE_{y_1,y_0}(\hat{y}|y_0)$) The expression $ctf\text{-}IE_{y_1,y_0}(\hat{y}|y_0,\mathbf{w},\mathbf{z})$ is a more fine-grained measure of indirect effect than $ctf\text{-}IE_{y_1,y_0}(\hat{y}|y_0)$, as the former captures more nuances of effects than the latter one. Whenever $ctf\text{-}IE_{y_1,y_0}(\hat{y}|y_0,\mathbf{w},\mathbf{z})$ is zero, $ctf\text{-}IE_{y_1,y_0}(\hat{y}|y_0)$ is zero. The converse is not true.*

*Proof.* We can further extend the total variation formula by the total probability theorem, and we arrive at

$$TV_{y_0,y_1}(\hat{y}) = \sum_{\mathbf{w},\mathbf{z}} ctf\text{-}DE_{y_0,y_1}(\hat{y}|y_0,\mathbf{w},\mathbf{z})P(\mathbf{w},\mathbf{z}|y_0) \tag{22}$$

$$- \sum_{\mathbf{w},\mathbf{z}} ctf\text{-}IE_{y_1,y_0}(\hat{y}|y_0,\mathbf{w},\mathbf{z})P(\mathbf{w},\mathbf{z}|y_0) \tag{23}$$

$$- ctf\text{-}SE_{y_1,y_0}(\hat{y}) \tag{24}$$

It is therefore easy to see that $ctf\text{-}IE_{y_1,y_0}(\hat{y}|y_0,\mathbf{w},\mathbf{z}) = 0 \implies ctf\text{-}IE_{y_1,y_0}(\hat{y}|y_0) = 0$, but there converse is not true in general. Proof of the falsity of the converse can be given by simple counterexamples. An interested reader can find some in Plecko & Bareinboim (2022). □

## B.2 JUSTIFICATION CONDITIONAL INDEPENDENCE AS CONSTRAINT

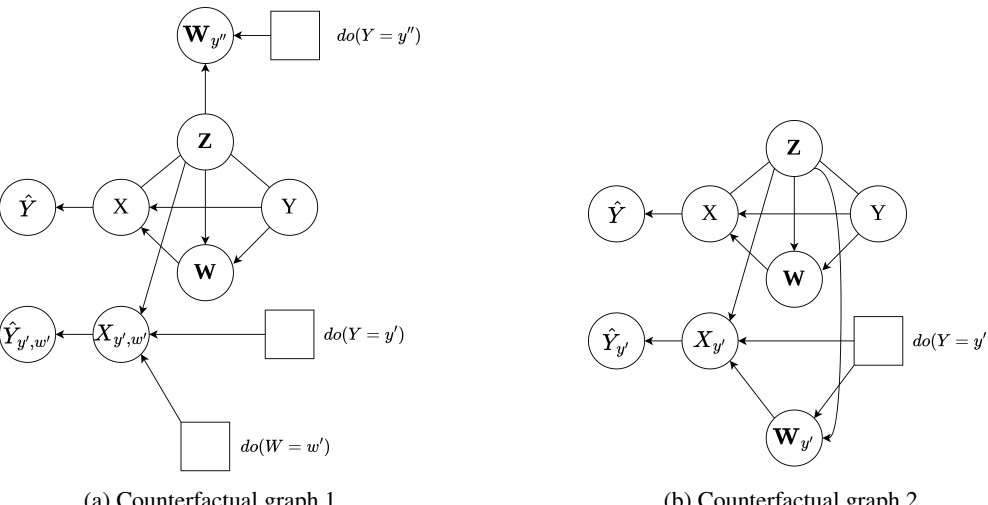

(a) Counterfactual graph 1          (b) Counterfactual graph 2

Figure B.3: Counterfactual Graphs needed for the proofs of propositions 2 and 3.

**Proposition 2 (Conditional Independence for unbiased Predictors)** SAM is given. Assume we have a prediction model $g_\theta$ that maps from the input $X$ to our target. Let's treat the predictions of our model as a random variable called $\hat{Y}$. If the prediction satisfies $\hat{Y} \perp \mathbf{W}, \mathbf{Z} \mid Y$, then $ctf\text{-}IE_{y_1,y_0}(\hat{y} \mid y) = ctf\text{-}SE_{y_1,y_0}(\hat{y}) = 0$, for any $y, y_0, y_1, \hat{y}$ in their ranges. The conditional independence criterion even ensures a stricter criterion, namely, $\hat{Y} \perp \mathbf{W}, \mathbf{Z} \mid Y \implies ctf\text{-}IE_{y_1,y_0}(\hat{y} \mid y, \mathbf{z}, \mathbf{w}) = 0$, for any $y, y_0, y_1, \hat{y}, \mathbf{w}, \mathbf{z}$ in the ranges of their respective random variables.

Outline

- Prop. 4 tells us that $ctf\text{-}IE_{y_1,y_0}(\hat{y}|y_0,\mathbf{w},\mathbf{z})$ is stronger than $ctf\text{-}IE_{y_1,y_0}(\hat{y}|y_0)$. Therefore, we will show the sufficiency of the stronger case, which will automatically apply to the weaker one.

- We need to derive some (in)dependence criteria based on the graph. We will use the theory from twin graphs (parallel worlds graphs) Balke & Pearl (1994); Avin et al. (2005) and their extensions made in the make-cg algorithm Shpitser & Pearl (2012).

- Another useful tool from Correa et al. (2021) will be used. We will refer to their Theorem 1 as counterfactual unnesting (*ctf-unnesting*) when we use it.

- We first show $\hat{Y} \perp \mathbf{W}, \mathbf{Z}|Y \implies ctf\text{-}IE_{y_1,y_0}(\hat{y}|y,\mathbf{z},\mathbf{w}) = 0$

- We afterwards show $\hat{Y} \perp \mathbf{W}, \mathbf{Z}|Y \implies ctf\text{-}SE_{y_1,y_0}(\hat{y}) = 0$

We begin with the graphical independence criteria. Relevant interventions and observations are in this case $\gamma = (\hat{Y}_{y',\mathbf{w}'}, \mathbf{w}_{y''}, y, \mathbf{w}, \mathbf{z})$. From the graph in Fig. B.3, we can conclude that $\hat{Y}_{y',\mathbf{w}'} \perp \mathbf{W}_{y''}, \mathbf{W}, Y \mid \mathbf{Z}$. Having these relations, we can prove that $\hat{Y} \perp \mathbf{W}, \mathbf{Z}|Y \implies ctf\text{-}IE_{y_1,y_0}(\hat{y}|y,\mathbf{z},\mathbf{w}) = 0$.

*Proof.*

$$
\begin{aligned}
ctf\text{-}IE_{y_1,y_0}(\hat{y}|y,\mathbf{w},\mathbf{z}) &= P(\hat{y}_{y_1,\mathbf{w}_{y_0}}|y,\mathbf{z},\mathbf{w}) - P(\hat{y}_{y_1}|y,\mathbf{z},\mathbf{w}) \\
&= P(\hat{y}_{y_1,\mathbf{w}_{y_0}}|y,\mathbf{z},\mathbf{w}) - P(\hat{y}_{y_1,\mathbf{w}_{y_1}}|y,\mathbf{z},\mathbf{w}) \\
&= \sum_{\mathbf{w}'} P(\hat{y}_{y_1,\mathbf{w}'}, \mathbf{W}_{y_0} = \mathbf{w}'|y,\mathbf{z},\mathbf{w}) \\
&\quad - \sum_{\mathbf{w}''} P(\hat{y}_{y_1,\mathbf{w}''}, \mathbf{W}_{y_1} = \mathbf{w}''|y,\mathbf{z},\mathbf{w}) && \textit{ctf-unnesting} \\
&= \sum_{\mathbf{w}'} P(\hat{y}_{y_1,\mathbf{w}'}|y,\mathbf{z},\mathbf{w}) P(\mathbf{W}_{y_0} = \mathbf{w}'|y,\mathbf{z},\mathbf{w}) \\
&\quad - \sum_{\mathbf{w}''} P(\hat{y}_{y_1,\mathbf{w}''}|y,\mathbf{z},\mathbf{w}) P(\mathbf{W}_{y_1} = \mathbf{w}''|y,\mathbf{z},\mathbf{w}) && \hat{Y}_{y',\mathbf{w}'} \perp \mathbf{W}_{y''} \mid \mathbf{Z} \\
&= \sum_{\mathbf{w}'} P(\hat{y}_{y_1,\mathbf{w}'}|y_1,\mathbf{z},\mathbf{w}') P(\mathbf{w}'|y_0,\mathbf{z}) && \hat{Y}_{y',\mathbf{w}'} \perp Y, \mathbf{W} \mid \mathbf{Z}, \\
&\quad - \sum_{\mathbf{w}''} P(\hat{y}_{y_1,\mathbf{w}''}|y_1,\mathbf{z},\mathbf{w}'') P(\mathbf{w}''|y_1,\mathbf{z}) && \mathbf{W}_{y''} \perp Y, \mathbf{W} \mid \mathbf{Z} \\
&= \sum_{\mathbf{w}'} P(\hat{y}|y_1,\mathbf{z},\mathbf{w}') P(\mathbf{w}'|y_0,\mathbf{z}) \\
&\quad - \sum_{\mathbf{w}''} P(\hat{y}|y_1,\mathbf{z},\mathbf{w}'') P(\mathbf{w}''|y_1,\mathbf{z}) \\
&= P(\hat{y}|y_1) \sum_{\mathbf{w}'} P(\mathbf{w}'|y_0,\mathbf{z}) && \hat{Y} \perp \mathbf{W}, \mathbf{Z}|Y \\
&\quad - P(\hat{y}|y_1) \sum_{\mathbf{w}''} P(\mathbf{w}''|y_1,\mathbf{z}) \\
&= 0
\end{aligned}
$$

$\square$

Next, we prove that $\hat{Y} \perp \mathbf{W}, \mathbf{Z}|Y \implies ctf\text{-}SE_{y_1,y_0}(\hat{y}) = 0$.

*Proof.*

$$
\begin{aligned}
SE_{y_1,y_0}(\hat{y}) &= P(\hat{y}_{y_1} \mid y_0) - P(\hat{y}_{y_1} \mid y_1) \\
&= \sum_{\mathbf{z}} [P(\hat{y}_{y_1} \mid y_0,\mathbf{z})P(\mathbf{z} \mid y_0) - P(\hat{y}_{y_1} \mid y_1,\mathbf{z})P(\mathbf{z} \mid y_1)] && \text{Law of total Prob.} \\
&= \sum_{\mathbf{z}} [P(\hat{y} \mid y_1,\mathbf{z})P(\mathbf{z} \mid y_0) - P(\hat{y} \mid y_1,\mathbf{z})P(\mathbf{z} \mid y_1)] && \hat{Y}_{y_1} \perp Y \mid \mathbf{Z} \\
&= P(\hat{y} \mid y_1) \sum_{\mathbf{z}} P(\mathbf{z} \mid y_0) - P(\hat{y} \mid y_1) \sum_{\mathbf{z}} P(\mathbf{z} \mid y_1) && \hat{Y} \perp \mathbf{W}, \mathbf{Z} \mid Y \\
&= 0
\end{aligned}
$$

$\square$

## B.3 CONSTANT DIRECT EFFECTS GIVEN Y,Y'

**Proposition** 3 (Constant Direct Effect for any $\mathbf{w}, \mathbf{z}$) The graph $\mathcal{G}$ and all implied assumptions are given. Assume we have a predictor $h_{\theta_h} \circ g_{\theta_g}$ and its output represented as a random variable $\hat{Y}$, such that $\hat{Y} \perp \mathbf{W}, \mathbf{Z} \mid Y$ holds. This predictor will have a constant direct effect for fixed $y_0, y_1$, for any $\mathbf{w}, \mathbf{z}$. This is compactly expressed as $ctf\text{-}DE_{y_0,y_1}(\hat{y}|y, \mathbf{w}, \mathbf{z}) = ctf\text{-}DE_{y_0,y_1}(\hat{y}|y, \mathbf{w}', \mathbf{z}')$, for any $y_0, y_1, \mathbf{w}, \mathbf{z}, \mathbf{w}', \mathbf{z}'$.

*Proof.*

$$
\begin{aligned}
ctf\text{-}DE_{y_1,y_0}(\hat{y}|y, \mathbf{w}, \mathbf{z}) &= P(\hat{y}_{y_1, \mathbf{w}_{y_0}}|y, \mathbf{z}, \mathbf{w}) - P(\hat{y}_{y_0}|y, \mathbf{z}, \mathbf{w}) \\
&= P(\hat{y}_{y_1, \mathbf{w}_{y_0}}|y, \mathbf{z}, \mathbf{w}) - P(\hat{y}_{y_0, \mathbf{w}_{y_0}}|y, \mathbf{z}, \mathbf{w}) \\
&= \sum_{\mathbf{w}'}[P(\hat{y}_{y_1, \mathbf{w}'}, \mathbf{W}_{y_0} = \mathbf{w}'|y, \mathbf{z}, \mathbf{w}) \\
&\quad - P(\hat{y}_{y_0, \mathbf{w}'}, \mathbf{W}_{y_0} = \mathbf{w}'|y, \mathbf{z}, \mathbf{w})] && \textit{ctf-unnesting} \\
&= \sum_{\mathbf{w}'}[P(\hat{y}_{y_1, \mathbf{w}'}|y, \mathbf{z}, \mathbf{w}) && \mathbf{W}_{y''} \perp Y, \mathbf{W} \mid \mathbf{Z} \\
&\quad - P(\hat{y}_{y_0, \mathbf{w}'}|y, \mathbf{z}, \mathbf{w})]P(\mathbf{W}_{y_0} = \mathbf{w}'|y, \mathbf{z}) \\
&= \sum_{\mathbf{w}'}[P(\hat{y}|y_1, \mathbf{z}, \mathbf{w}') && \hat{Y}_{y', \mathbf{w}'} \perp Y, \mathbf{W} \mid \mathbf{Z} \\
&\quad - P(\hat{y}|y_0, \mathbf{z}, \mathbf{w}')]P(\mathbf{W}_{y_0} = \mathbf{w}'|y, \mathbf{z}) \\
&= [P(\hat{y}|y_1) - P(\hat{y}|y_0)]\sum_{\mathbf{w}'} P(\mathbf{W}_{y_0} = \mathbf{w}'|y, \mathbf{z}) && \hat{Y} \perp \mathbf{W}, \mathbf{Z} \mid Y \\
&= P(\hat{y}|y_1) - P(\hat{y}|y_0)
\end{aligned}
$$

$\square$

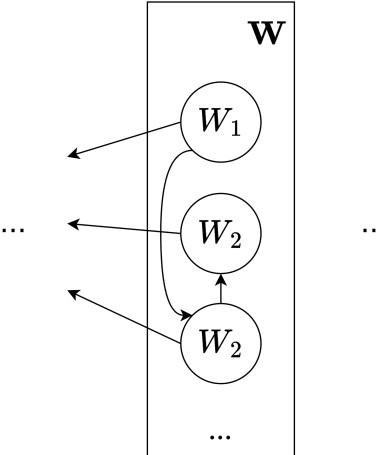

Figure B.4: Allowing edges between nodes in $\mathbf{W}$.

## B.4 CHANGING ASSUMPTIONS IN SAM

**Retaining some intermediate effects** In our framework, we made assumptions by introducing the graph $\mathcal{G}$. We further assumed that the only relevant effects are direct. Not all intermediate effects, however, might be undesirable in general. In this case, we can exclude variables $\mathbf{W}' \in \mathbf{W}$ in the expression $\hat{Y} \perp \mathbf{W} \setminus \mathbf{W}', \mathbf{Z}|Y$, if we want to retain the mediated effects of $\mathbf{W}'$ on $\hat{Y}$, besides the direct effects of $Y$ on $\hat{Y}$.

**Connecting nodes in W**    We initially restricted the arrows in the mediator set $\mathbf{W}$ by not allowing any connections between two distinct nodes $W_1 \in \mathbf{W}$ and $W_2 \in \mathbf{W}$. As seen in Fig. B.4, we can allow intra-connections between variables in $\mathbf{W}$. But then, retaining intermediate effects of a subgroup of nodes $\mathbf{W}' \subset \mathbf{W}$ will be non-trivial and depend on the actual connections that remain.

**Simplifying The Graph**    Dropping edges and nodes is always possible, and the sufficiency of our conditional independence criterion $\hat{Y} \perp \mathbf{W}, \mathbf{Z} \mid Y$ for the Prop. 2 and Prop. 3 stay true in these cases.

## C    DISTANCE CORRELATION

### C.1    PROOFS OF LEMMATA 1 AND 2

*Proof of Lemma 1.*  To prove Lemma 1, either the steps of the original unconditional distance correlation paper Székely et al. (2007) can be followed, as the proof is analog in this case, or the steps in Wang et al. (2015) can be followed. □

*Proof of Lemma 2.*  The proof for the conditional characteristic function in our Lemma 2 is similar to the proof of Lemma 1 in Székely et al. (2007) for the unconditional characteristic function. Considering that the weight function $w$ is defined in Sec.3, the same steps as in Székely et al. (2007) can be applied in the proof. □

## D EXPERIMENTS

### D.1 EXPERIMENTAL SETUP AND JUSTIFICATION

We acknowledge the big challenge of model selection in domain generalization, bias mitigation, fairness, and similar contexts. Regular model selection strategies, such as simply choosing the best validation target loss (MSE), are not applicable, as we would choose models that would have overfitted on the in-domain data and, thus, rely heavily on the bias.

Since all the methods we evaluate include a regularization term, we use the total loss on the in-domain validation set instead. This measure provides a trade-off between invariance and performance by including the mean squared error (MSE) alongside the invariance penalty.

Our next decision was inspired by Pogodin et al. (2022) to plot the OOD-test MSEs for different regularization parameters ($\lambda$). The reason for doing so is to better grasp how different regularization strengths influence OOD performance for a better comparison of regularization effects on a pure algorithmic level.

### D.2 EXTENDED YALEB

We follow Yale (2001) to create the Extended Yale B dataset from the official *Yale Face Database B* Georghiades et al. (2001), available at `http://cvc.cs.yale.edu/cvc/projects/yalefacesB/yalefacesB.html`.

Table 1: Parameter Settings for YaleB Dataset

| Method | Parameters |
|--------|-----------|
| Circe | LR: [0.01, 0.001, 0.0001] |
| | sigma_Y: [1.0, 0.1, 0.01, 0.001] |
| | ridge_lambda: [0.01, 0.1, 1.0, 10.0, 100.0] |
| | sigma_Z: 0.01 |
| | sigma_pred: 0.01 |
| | lambda: [1.0, 10.0, 100.0, 1000.0] |
| HSCIC | LR: [0.01, 0.001, 0.0001] |
| | sigma_Y: [1.0, 0.1, 0.01, 0.001] |
| | ridge_lambda: [0.01, 0.1, 1.0, 10.0, 100.0] |
| | sigma_Z: 0.01 |
| | sigma_pred: 0.01 |
| | lambda: [1.0, 10.0, 100.0, 1000.0] |
| DISCO | LR: [0.01, 0.001, 0.0001] |
| | sigma_Y: 0.01 |
| | lambda: [1.0, 2.0, 5.0, 10.0, 100.0] |

## D.3 DSPRITES

Table 2: Parameter Settings for dSprites Dataset

| Method | Parameters |
|---|---|
| Circe | LR: [0.001, 0.0001]
sigma_Y: [1.0, 0.1, 0.01, 0.001]
ridge_lambda: [0.01, 0.1, 1.0, 10.0, 100.0]
sigma_Z: 0.01
sigma_pred: 0.01
lambda: [1.0, 10.0, 100.0, 1000.0] |
| HSCIC | LR: [0.001, 0.0001]
sigma_Y: [1.0, 0.1, 0.01, 0.001]
ridge_lambda: [0.01, 0.1, 1.0, 10.0, 100.0]
sigma_Z: 1.0
sigma_pred: 0.01
lambda: [1.0, 10.0, 100.0, 1000.0] |
| DISCO | LR: [0.001, 0.0001]
sigma_Y: 0.01
lambda: [1.0, 2.0, 5.0, 10.0, 100.0] |

dSprites Matthey et al. (2017) images were generated using the LOVE framework, which is licensed under zlib/libpng license. More information is available at https://github.com/google-deepmind/dsprites-dataset.

## D.4 3D SIMULATION

Table 3: Parameter Settings for 3D Simulation Dataset

| Method | Parameters |
|---|---|
| Circe | LR: [0.0001]
sigma_Y: [1.0, 0.1, 0.01, 0.001]
ridge_lambda: [0.01, 0.1, 1.0, 10.0, 100.0]
sigma_Z: [0.01, 0.1, 1.0, 10.0]
sigma_pred: [0.01, 0.1, 1.0, 10.0]
lambda: [1.0, 10.0, 20.0, 100.0, 1000.0, 10000.0] |
| HSCIC | LR: [0.0001]
sigma_Y: [1.0, 0.1, 0.01, 0.001]
ridge_lambda: [0.01, 0.1, 1.0, 10.0, 100.0]
sigma_Z: [0.01, 0.1, 1.0, 10.0]
sigma_pred: [0.01, 0.1, 1.0, 10.0]
lambda: [1.0, 10.0, 20.0, 100.0, 1000.0, 10000.0] |
| DISCO | LR: [0.0001]
sigma_Y: [0.1, 1.0, 10.0]
lambda: [1.0, 2.0, 5.0, 10.0, 100.0] |

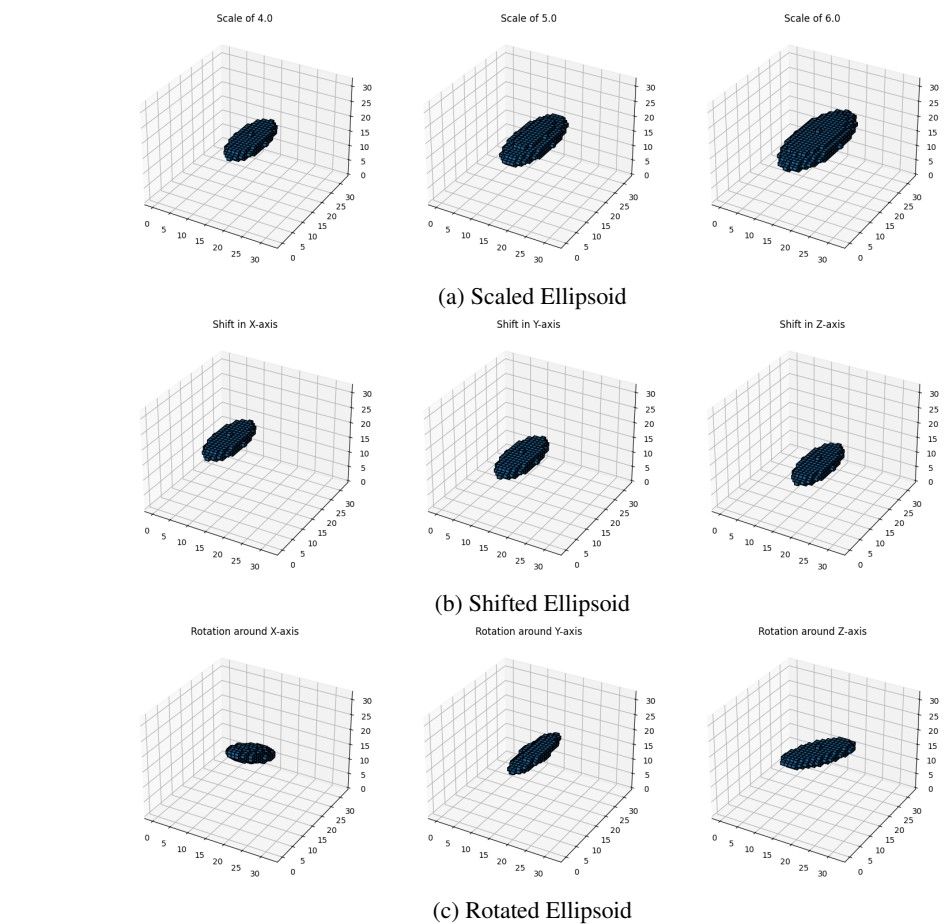

(a) Scaled Ellipsoid

(b) Shifted Ellipsoid

(c) Rotated Ellipsoid

Figure D.5: Illustration of ellipsoids from our 3D experiment. Note that these are voxelized versions of it. We also add Gaussian noise and assign brightness to each ellipsoid as another source of distraction.

**Structural Equations in-domain**    For the 3D ellipsoid data, $X$ are the $32^3$ image cubes with an ellipsoid placed into them, $Y$ is the target (the same rotation angle for all three axes); $S$ is the scale factor that determines the overall size of the ellipsoids; $C$ is the three-dimensional random vector for the ellipsoid centers; and $B$ is the brightness of the ellipsoid. Note that we fixed the ellipsoids scale ratios along all axes in cartesian space to 1.0:2.5:0.5. Structural equations for the 3D dataset are below.

$$Y \sim \mathcal{N}(0, \left(\frac{\pi}{4}\right)^2) \quad S = 4 + \sin(y) + \epsilon_S \tag{25}$$

$$C = \begin{bmatrix} 16 \\ 16 \\ 16 \end{bmatrix} + 1.5 \times \sin(y) \begin{bmatrix} 1 \\ 1 \\ 1 \end{bmatrix} + \tag{26}$$

$$B = \exp(\sin(y) \times 1.25) + \exp(\epsilon_B) + 1 \tag{27}$$

$$X = \text{Ellipsoid}_{Y,S,C,B} + \epsilon_X \tag{28}$$

$$\epsilon_C = \mathcal{N}(\mathbf{0}, (1.1)^2\mathbf{I}) \quad \epsilon_S \sim \text{Uniform}(0, 1.5) \quad \epsilon_B = \mathcal{N}(0, 0.5) \tag{29}$$

$$\epsilon_X \sim \mathcal{N}(0, (0.1)^2) \tag{30}$$

The procedure for the 3D ellipsoids embedded in the 3D cubes which will together form the images, is given by

1. Define a 3D grid of coordinates $(x, y, z)$ for an image of size $N$:

$$x, y, z = \text{meshgrid}\left(\text{linspace}(0, N-1, N)^3\right)$$

2. Compute rotation matrices for each axis:

$$R_x = \begin{bmatrix} 1 & 0 & 0 \\ 0 & \cos(\alpha) & -\sin(\alpha) \\ 0 & \sin(\alpha) & \cos(\alpha) \end{bmatrix}, \quad R_y = \begin{bmatrix} \cos(\beta) & 0 & \sin(\beta) \\ 0 & 1 & 0 \\ -\sin(\beta) & 0 & \cos(\beta) \end{bmatrix}$$

$$R_z = \begin{bmatrix} \cos(\gamma) & -\sin(\gamma) & 0 \\ \sin(\gamma) & \cos(\gamma) & 0 \\ 0 & 0 & 1 \end{bmatrix}$$

$$R = R_z R_y R_x$$

3. Adjust coordinates and apply rotation:

$$\text{xyz} = \begin{bmatrix} x - \text{centers}_x \\ y - \text{centers}_y \\ z - \text{centers}_z \end{bmatrix}$$

$$\text{rotated\_xyz} = R \cdot \text{xyz}$$

4. Apply the ellipsoid equation:

$$\text{ellipsoid} = \left(\frac{\text{rotated\_xyz}_x}{\text{radii}_x}\right)^2 + \left(\frac{\text{rotated\_xyz}_y}{\text{radii}_y}\right)^2 + \left(\frac{\text{rotated\_xyz}_z}{\text{radii}_z}\right)^2 \leq 1$$

**Structural Equations OOD - Interventional World**

$$S = \text{Uniform}(0, 4) + 3 \tag{31}$$

$$C = \begin{bmatrix} 16 \\ 16 \\ 16 \end{bmatrix} + \mathcal{N}(\mathbf{0}, (1.5)^2 \mathbf{I}) \tag{32}$$

$$B = \text{Uniform}(0, 2.0) + 2.0. \tag{33}$$

## D.5    ADDITIONAL MATERIAL

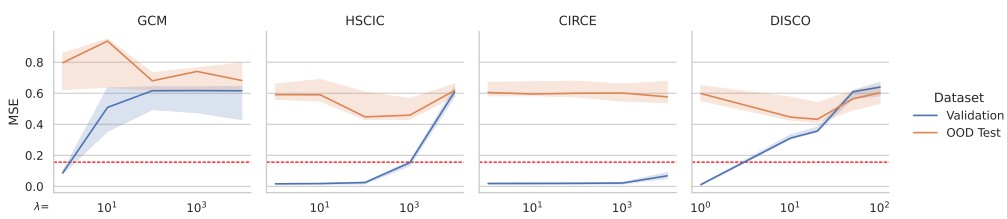

Figure D.6: Own 3D simulation experiment including GCM as a benchmark. It confirms the findings of Pogodin et al. (2022) that GCM is the weakest method.

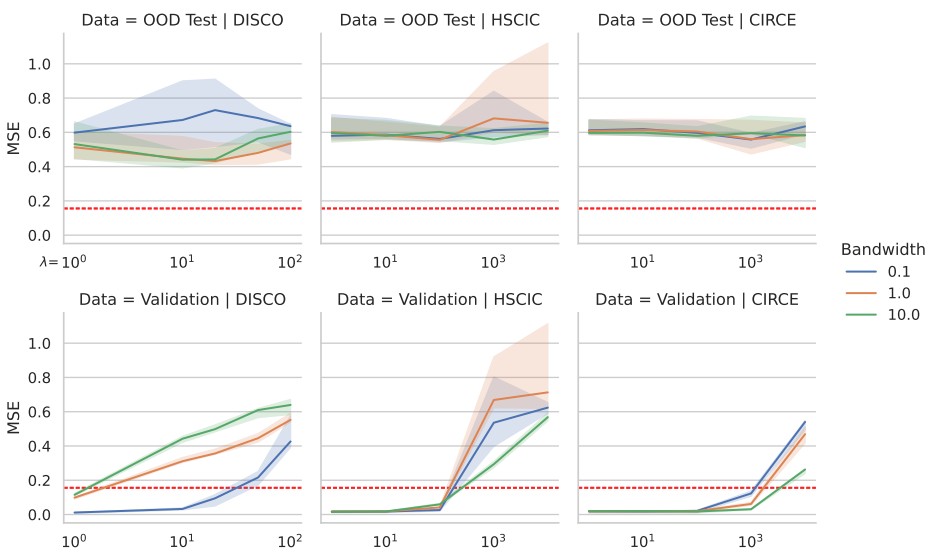

Figure D.7: In this experiment, we fixed the bandwidth parameter sigma_pred of HSCIC and CIRCE to 0.01 for our simulated dataset and plot the results for different values of sigma_Z as Bandwidth. For DISCO, Bandwidth corresponds to sigma_Y. All methods show a sensitivity to the selected bandwidth parameters. While DISCO only has a single one of them, the other methods have 3 of these, rendering them more sensitive to model choice mechanisms and similar. Note we randomly fixed the remaining bandwidth hyperparameters for HSCIC and CIRCE. We can easily find even more sensitive bandwidth combinations that make these methods more volatile.

