# OpenReview forum: "DISCO: Mitigating Bias in Deep Learning with Conditional DIStance COrrelation"
_ICLR.cc/2025/Conference — Submitted to ICLR 2025_

### Official Review · Reviewer_dHmH · 2024-11-02

**Soundness:** 3
**Presentation:** 2
**Contribution:** 2
**Rating:** 5
**Confidence:** 3

**Summary:**

The paper introduces a conditional independence criterion aimed at eliminating bias in deep learning models by ensuring that predictions rely exclusively on task-relevant information paths. It then designs a regularization term to implement this criterion, preventing models from depending on irrelevant or spurious variables. Several experiments are conducted to validate the proposed method’s effectiveness.

**Strengths:**

1. The paper is clearly written, and it successfully reduces the number of hyperparameters required for conditional independence regularization.
2. Theoretical derivations are detailed and comprehensive, demonstrating a strong understanding of causal inference principles.

**Weaknesses:**

1. The algorithm incorporates elements from previous work without sufficient clarity, and the lack of detail makes some parts challenging to follow.
2. Experimental results are less compelling than the theoretical contributions suggest. MSE improvements are marginal, and the variance on the dSprites dataset is noticeably higher than the baseline.

**Questions:**

1. The section on SCM appears somewhat disconnected from later sections. Could you clarify its relevance to the overall framework?
2. The experiments focus primarily on computer vision simulation datasets. Could you test the method on additional settings to demonstrate its broader applicability?
3. Given DISCO’s computational complexity of $O(b^3)$, could you provide details on the batch size settings in your experiments and include timing comparisons with baseline methods?

---

### Official Review · Reviewer_aiwF · 2024-11-03

**Soundness:** 2
**Presentation:** 2
**Contribution:** 2
**Rating:** 3
**Confidence:** 3

**Summary:**

This paper introduces a formalization of ensuring the robustness of machine learning model predictions from via an anti-causal formulation.  They formalize a standard anticausal model (SAM) and prove that a predictor that satisfies a particular conditional independence relationship in this setting will be guaranteed to learn a stable prediction, invariant to spurious information. Although explicitly regularizing for this independence condition is challenging, especially in a black-box deep learning setting, they introduce a conditional  distance correlation based method -- which can be introduced as a simple regularization term -- to impose the desired conditional independence. While this regularization cannot be calculated in closed form estimator, they introduce a kernel regression method for estimating the quantity. The proposed regularization term is tested empirically on a set of 2d and 3d image tasks.

**Strengths:**

This paper tackles an important problem at the intersection of fairness and robustness in machine learning systems. They propose a general formalization of these tasks and, inspired by it, propose a novel regularization term. They explain some limitations of previous methods to ensure stable or invariant classification, such as the inability to handle continuous confounding variables, classification tasks, and the necessity for stratification.

**Weaknesses:**

The experimental results seem to be limited. Firstly, the method is only tested on mostly simulation-based datasets, although there also exist many real-world datasets for spurious correlations. It would also be helpful for them to compare the method on some settings where there is a single, categorical spurious variable-- as there are extensive baseline methods in those settings (i.e. Sagawa et.al) . Currently, the method is compared to a relatively small number of baselines. Moreover, I may be missing something from my reading of the plots, but it seems that the gains of the method over the limited baselines that are tested are small and unlikely to be statistically significant. Maybe the authors could provide some additional statements explaining why simply being competitive with existing methods (I..e CIRCE) represents an important contribution. In particular, as the authors themselves acknowledge, DISCO is computationally more expensive which calls the contribution and relevance of the paper into question. I think the authors could better highlight under what settings they would expect DISCO to achieve significantly superior performance and ideally they could find some way to empirically shown this.

**Questions:**

Please respond to the points raised in Weaknesses.

---

### Official Review · Reviewer_M1Wq · 2024-11-03

**Soundness:** 4
**Presentation:** 4
**Contribution:** 4
**Rating:** 8
**Confidence:** 2

**Summary:**

This paper first proposes an anti-causal model to predict targets Y from input X considering possible causal, non-causal pathways. Theoretical result is derived using total variation to show that if the prediction model satisfies a conditional independence result given targets Y, its predictions will be unbiased from variables with undesired effects - Z,W due to spurious or irrelevant information. Further, the paper shows that the prediction model can be pushed in this direction through an additional regularization term during training. Distance correlation is used to this effect in deep learning models - using model features, data distractors and targets to formulate the regularization term. While some prior work also uses regularization for conditional independence, this method allows achieving competitive results to the baselines efficiently with lesser hyperparameters. This method extends beyond classification to continuous labels/vector.

**Strengths:**

1. The paper is the first to formulate the use of distance correlation for learning causal representations with deep learning networks.
2. I find the paper to be mostly well written with ample pointers to underlying concepts and prior work.
3. Experiments are shown to compete with baseline methods CIRCE, HSCIC across different types of domain changes - human image data with different illuminations, 2D shape data, 3D simulated data. Experiments are clearly described and compared to baseline.
4. An anonymized repository of well organized code is provided. This helped in clearly mapping the theoretical result to its application on mentioned datasets.

**Weaknesses:**

Perhaps more discussion and analysis with visualizations of model’s failure cases could help in understand the shortcomings of this method other than computation cost.

**Questions:**

Other than anti causal scenario proposed, what are some other frameworks where the authors think conditional distance correlation can be used and the field should pursue?

---

### Official Review · Reviewer_WAJ2 · 2024-11-04

**Soundness:** 3
**Presentation:** 2
**Contribution:** 2
**Rating:** 3
**Confidence:** 3

**Summary:**

The authors propose a method that decreases reliance on irrelevant features by optimizing for conditional independence using conditional distance correlation for regression tasks. Through the Standard Anti-Causal Model (SAM) framework, the authors analyze and address pathways that could introduce bias, and encourage the model to focus solely on the direct relationship between input and target. Experiments on two datasets demonstrate that DISCO matches or slightly outperforms two other bias-mitigation methods.

**Strengths:**

- The paper rigorously shows that enforcing cdCor minimizes conditional dependence, demonstrated through Proposition 2 and Proposition 3, and proves that when Y hat ⊥W, Z ∣ Y holds, indirect and spurious effects are null.

- DISCO addresses a gap in the literature by focusing on bias mitigation for regression, and expands fairness techniques beyond their typical focus on classification tasks.

**Weaknesses:**

- The introduction of the paper is not very well-written — it includes literature review, but it does not provide an example of real-world tasks that DISCO tackles or this paper's motivation.

- In contributions on page two it’s mentioned that SAM is a generalization of the standard fairness models. Can the authors explain how SAM is different from more recent models proposed? e.g. multi-domain shift in figure 1 from this paper: https://proceedings.mlr.press/v238/tsai24b/tsai24b.pdf

- Related Work on Conditional Independence Regularization: this paragraph should be moved from section 3 to related work.

**Questions:**

- Can the authors summarize the results of their experiments in a table, and compare them with baselines? Have the authors done a full literature review on Domain Generalization for regression? It seems like some baselines such as are missing. e.g. https://arxiv.org/abs/2312.17463v1

- Can the authors add more regression tasks that have been used in the DG literature more recently? such as the PovertyMap dataset from WILDS?

---

### Meta-Review · Area_Chair_eb4n · 2024-12-20

**Metareview:**

The authors did not provide responses to the reviews, and several major concerns including unclear motivation, unclear differences with existing methods, and absence of baselines or datasets remain unaddressed. As a result, we recommend rejecting this paper.

**Additional Comments On Reviewer Discussion:**

The reviewers raised several major concerns including unclear motivation, unclear differences with existing methods, and the absence of baselines or datasets.  The authors did not provide responses to the reviews.

---

### Decision · Program_Chairs · 2025-01-22

Reject